# Therapeutic B-cell depletion reverses progression of Alzheimer's disease

Ki Kim [1,10], Xin Wang[1,10], Emeline Ragonnaud[1,10], Monica Bodogai[1,10], Tomer Illouz [2,3,4], Marisa DeLuca [1], Ross A. McDevitt[5], Fedor Gusev[6], Eitan Okun [2,3,4], Evgeny Rogaev[6,7,8,9] & Arya Biragyn [1✉]

The function of B cells in Alzheimer's disease (AD) is not fully understood. While immunoglobulins that target amyloid beta (Aβ) may interfere with plaque formation and hence progression of the disease, B cells may contribute beyond merely producing immunoglobulins. Here we show that AD is associated with accumulation of activated B cells in circulation, and with infiltration of B cells into the brain parenchyma, resulting in immunoglobulin deposits around Aβ plaques. Using three different murine transgenic models, we provide counterintuitive evidence that the AD progression requires B cells. Despite expression of the AD-fostering transgenes, the loss of B cells alone is sufficient to reduce Aβ plaque burden and disease-associated microglia. It reverses behavioral and memory deficits and restores TGFβ+ microglia, respectively. Moreover, therapeutic depletion of B cells at the onset of the disease retards AD progression in mice, suggesting that targeting B cells may also benefit AD patients.

[1] Immunoregulation Section, Laboratory of Immunology and Molecular Biology, National Institute on Aging, Baltimore, MD, USA. [2] The Mina and Everard Goodman faculty of Life Sciences, Ramat Gan, Israel. [3] The Gonda Brain Research Center, Bar Ilan University, Ramat Gan, Israel. [4] The Paul Feder Laboratory on Alzheimer's disease research, Bar Ilan University, Ramat Gan, Israel. [5] Mouse Phenotyping Unit, Comparative Medicine Section, National Institute on Aging, Baltimore, MD, USA. [6] Department of Genomics and Human Genetics, Institute of General Genetics, Russian Academy of Sciences, Moscow, Russia. [7] Center for Genetics and Genetic Technologies, Faculty of Biology, Faculty of Bioengineering and Bioinformatics, Lomonosov Moscow State University, Moscow, Russia. [8] Department of Psychiatry, University of Massachusetts Medical School, Worcester, MA, USA. [9] Sirius University of Science and Technology, Sochi, Russia. [10] These authors contributed equally: Ki Kim, Xin Wang, Emeline Ragonnaud, Monica Bodogai. ✉email: biragyna@mail.nih.gov

Alzheimer's disease (AD) is a progressive neurodegenerative disease that mostly affects elderly people. It is associated with impaired clearance of toxic protein aggregates from the brain parenchyma, such as amyloid-β (Aβ) peptides of aberrantly cleaved amyloid precursor protein (APP)[1]. Although resident microglial cells phagocytize extracellular Aβ plaques with the help of astrocytes and TGFβ (refs. [2,3]), chronic inflammation, and Aβ production dysregulate this process, causing proliferation and subsequent replacement of homeostatic microglia with disease-associated microglia (DAM)[4]. As in mice with genetic TGFβ deficiency, which suffer from microgliosis and neuronal death[5], DAM further exacerbate neuroinflammation and neuronal degeneration in AD[6,7], at least in part, through expression of proinflammatory cytokines and downregulation of phagocytosis of Aβ plaques[8–10]. Consistent with a positive association between AD risk and old age[11], when systemic inflammation is increased[12], disease progression also depends on peripheral inflammation and activation of innate immune cells[13]. The role of the adaptive immunity in AD however remains poorly understood, and is mostly linked to T cells exerting both beneficial and harmful functions. For example, in APP/PS1 mice expressing the K670N and M671L Swedish mutations of human APP and L166P mutated presenilin 1 (PS1), the infiltration of Th1 CD4+ T cells in the brain either improves[14] or exacerbates AD[15]. Moreover, the amelioration of AD in Rag2-deficient APP/PS1 mice is primarily linked to the loss of pathogenic T cells[16]. Although Rag deficiency also retards development of functional B cells (besides T cells), the role of B cells in AD is rarely explored and mostly considered to be beneficial. Their main product, nonspecific immunoglobulin and Aβ-specific antibodies in peripheral blood (PB) and cerebrospinal fluid (CSF) of people[17,18] and mice[19,20] with AD, provides neuroprotective benefit. In 5×FAD mice, which overexpress APP with the Swedish (K670N, M671L), Florida (I716V) and London (V717I) mutations, and PS1 with M146L and L286V mutations, Rag deficiency exacerbates AD due to loss of nonspecific immunoglobulin that activates microglial phagocytosis and consequent clearance of Aβ plaques[21]. However, B cells are a heterogenous population of cells. Their function and subset accumulation are regulated by the inflammatory milieu. For example, we recently reported that inflammaging activates monocytes to convert innate B1a cells into pathogenic 4-1BBL+ TNFα+ MHC-IHigh B cells (termed 4BL cells), which then induce cytolytic CD8+ T cells and insulin resistance in elderly humans, macaques, and mice[22,23]. However, the role of these or other activated B cells in aging-associated diseases, such as AD remains unknown.

Here, using three, widely utilized transgenic AD mouse models with or without genetic B-cell deficiency, we provide evidence and mechanistic insight that the loss of mature B cells alone is sufficient to markedly retard the AD progression. Although B cells are thought not to infiltrate the AD brain, we link their presence in the brain parenchyma to the disease progression. Importantly, therapeutic depletion of circulating B cells at the onset of the disease eliminates B cells, and their immunoglobulin deposits in the brain and blocks the AD manifestation. Our data not only warrants a new look to B cells as pathogenic, AD-promoting cells in humans with AD, but also suggests that their B cells should be targeted to control the disease progression.

## Results

To assess the involvement of B cells in AD, we first performed a flow cytometry evaluation of B cells in the circulation of congenic and sex- and age-matched C57BL/6 (WT) and 3×TgAD mice, a model for early-onset AD (EOAD, harbors the Swedish APP mutation, the M146V mutation on PS1 and the P301L in MAPT)[24,25]. Note: from here on and unless specified, we used

aged (60–70 weeks old) female 3×TgAD mice due to their more pronounced AD-like symptom manifestation[25]. Compared with control mice, 3×TgAD mice significantly upregulated the frequency and numbers of B cells in the circulation and secondary lymphoid organs (Supplementary Fig. 1A and not depicted; gating strategy is in Supplementary Fig. 9A), with innate CD5+ B1a and CD5− B1b cells (CD11b+ or/and CD23− CD43+ CD19+cells, respectively) markedly increased, and conventional B2 cells decreased in the spleen and cervical lymph nodes (cLN; Fig. 1A–C and Supplementary Fig. 1B–D). The 3×TgAD mice also contained higher amounts of activated B cells expressing IFNγ, IL6, IL10, and TGFβ in circulation than control mice (Fig. 1D–G), mostly within B1a cells and, at lesser extent B1b and follicular B cells (Supplementary Fig. 1E–H). The AD mouse B cells, particularly B1a cells, markedly upregulated 4-1BBL (Supplementary Fig. 1I, J), resembling pathogenic B1a cells of aged subjects[22,23] and suggesting their potential involvement in AD. To test this possibility, we have generated B-cell-deficient 3×TgAD mice (termed 3×TgAD-BKO, Supplementary Fig. 2A) by crossing 3×TgAD mice with JHT mice, which lack functional B cells due to the immunoglobulin JH locus deletion that terminates B-cell development at the pro-B cell stage[26]. When mice reached 50–60 weeks of age, they were evaluated for hippocampus-dependent cognitive behavior, using the Morris water maze (MWM) task. During the 5-day hidden platform training, both 3×TgAD and 3×TgAD-BKO mice showed comparable performance (not depicted). However, 24 h after the last training session, a probe trial conducted in the absence of the hidden platform, revealed that 3×TgAD-BKO mice spent a significantly longer time in the target quadrant than their 3×TgAD littermates ($p = 0.017$, $n = 13$–15, Fig. 2A), indicating a stronger spatial memory of the platform's location. We also tested exploratory behavior anomalies using the open field arena (OFA, a commonly used assay in 3×TgAD mice[27,28]). While 3×TgAD mice exhibited reduced activity as compared with WT mice in the OFA, the impairment was no longer detectable in age-matched 3×TgAD-BKO littermates (left panel, one-way ANOVA $F(2,29) = 21.68$, $p < 0.0001$; WT vs 3×TgAD $p < 0.0001$, 3×TgAD vs 3×TgAD-BKO $p < 0.0001$, $n = 10$–11, and Fig. 2B), implying that 3×TgAD-associated behavioral impairments required B cells.

To confirm this conclusion, we tested APP/PS1 mice, another model of EOAD, that exhibit earlier AD pathology compared with 3×TgAD mice[29]. Of note: subsequent experiments were therefore conducted in female and male, 20–35-week-old mice. Flow cytometry evaluation of circulating B cells revealed that APP/PS1 mice also markedly upregulated B cells expressing IL10 (Fig. 2C) as in 3×TgAD mice (Fig. 1F). However, APP/PS1 mice did not upregulate 4-1BBL+ B cells (presumably due to their relatively young age[22,23]) despite a significant increase of B1a cells in circulation and the cLN compared with control littermates (Supplementary Fig. 2B, C). We next generated B-cell-deficient APP/PS1-BKO mice by crossing APP/PS1 and JHT mice. Analysis of spatial learning using the MWM revealed that compared with age- and sex-matched APP/PS1 or WT littermates, B-cell deficiency significantly improved learning deficiency in APP/PS1 mice as evident by latency to reach the escape platform ($F(2,33) = 8.288$, $p = 0.0012$; WT vs APP/PS1 $p = 0.0006$, APP/PS1 vs APP/PS1-BKO $p = 0.003$, $n = 12$, Fig. 2D); and in the number of platform region crossings in a probe test conducted 24 h following learning (Kruscal–Wallis ANOVA $p = 0.0004$; WT vs APP/PS1 $p = 0.0002$; APP/PS1 vs APP/PS1-BKO $p = 0.028$, Fig. 2E). Consistent with the lack of exploratory behavior impairment in APP/PS1 mice[30], no difference in exploratory behavior of our mice in the OFA (not depicted). Thus, the spatial learning impairments exhibited by APP/PS1 are B-cell dependent.

Because the memory impairments in EOAD are caused by accumulation of Aβ plaques and hippocampal microgliosis[31], we

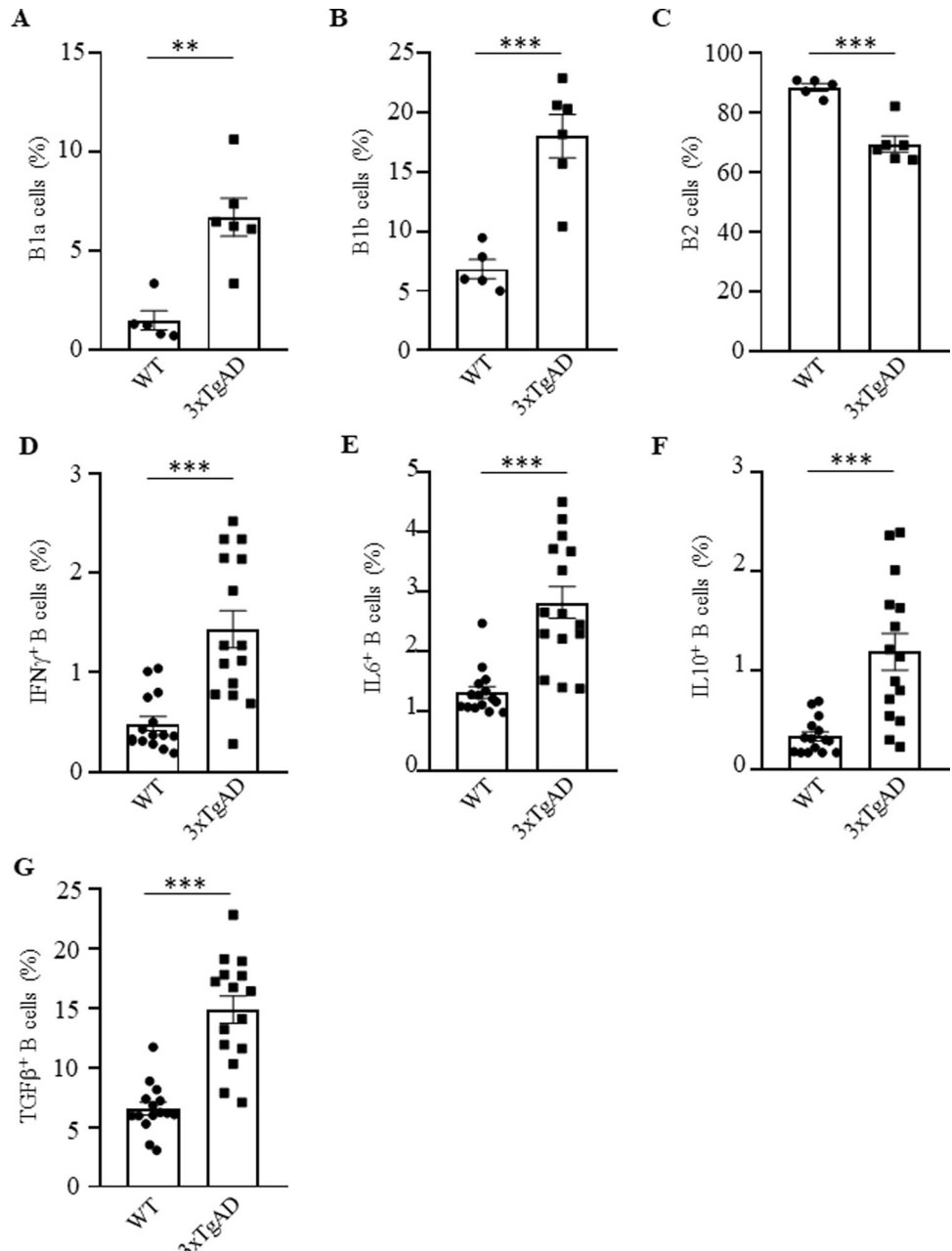

**Fig. 1 Activated B cells were increased in 3×TgAD mice.** Compared with congenic, age- and sex-matched WT mice, B1a (CD5+CD11b+CD19+, **A**) and B1b cells (CD5−CD11b+CD19+, **B**) were increased, while B2 cells (CD5−CD11b−CD19+, **C**) were decreased in the cervical lymph nodes of 3×TgAD mice. AD also activated B cells, as they upregulated IFNγ (**D**), IL6 (**E**), IL10 (**F**), and TGFβ (**G**) in the peripheral blood of mice. Frequency (%) mean ± SEM is shown; each symbol is for a single mouse, $n = 5$–6 in **A**–**C**, $n = 15$ in **D**–**G**. Gating strategy is shown in Supplementary Fig. 9A. **$p < 0.01$; ***$p < 0.001$ in unpaired $t$ test.

conducted an immunofluorescent analysis for Aβ plaques (using the 6E10 Ab) and ionized calcium binding adaptor 1 (Iba, a microglial cell marker) in cryopreserved brain sections of APP/PS1-BKO (30 weeks old), 3×TgAD-BKO (60–70 weeks old) and age- and sex-matched control mice. Given that in this model the early intraneuronal Aβ deposition in the subiculum is linked to cognitive impairments[32], and that the subiculum and hippocampal CA1 atrophy is increased in AD patients[33], from hereon we primarily analyzed the subiculum. It showed a marked upregulation of Aβ plaques in APP/PS1 and 3×TgAD mice (as compared with healthy control mice), which was reversed in APP/PS1-BKO and 3×TgAD-BKO mice respectively ($n = 4$–7, Fig. 3A–C and Supplementary Fig. 3A–D). Analysis of frontal

cortex and hippocampus in 3×TgAD-BKO mice also revealed that the increase of soluble Aβ$_{42}$ and Aβ$_{40}$ was reversed ($n = 2$–6, Fig. 3D and Supplementary Fig. 3E). Since both 3×TgAD and 3×TgAD-BKO mice express high levels of the APP transgene in hippocampal neurons (Fig. 3E), we concluded that the benefit of the B-cell deficiency was probably in reduced formation and/or increased clearance of Aβ peptides. Accordingly, large ameboid microglial cells (>5 μm$^2$), which indicate dysfunctional overactivation[34,35] and impaired clearance of Aβ plaques[36], were significantly decreased in 3×TgAD-BKO mice to almost that of in WT mice ($p < 0.05$, compared with 3×TgAD, $n = 3$–9, Fig. 3F and Supplementary Fig. 3F, G). In contrast, activated microglia (regardless of their size) remained increased in APP/PS1 and

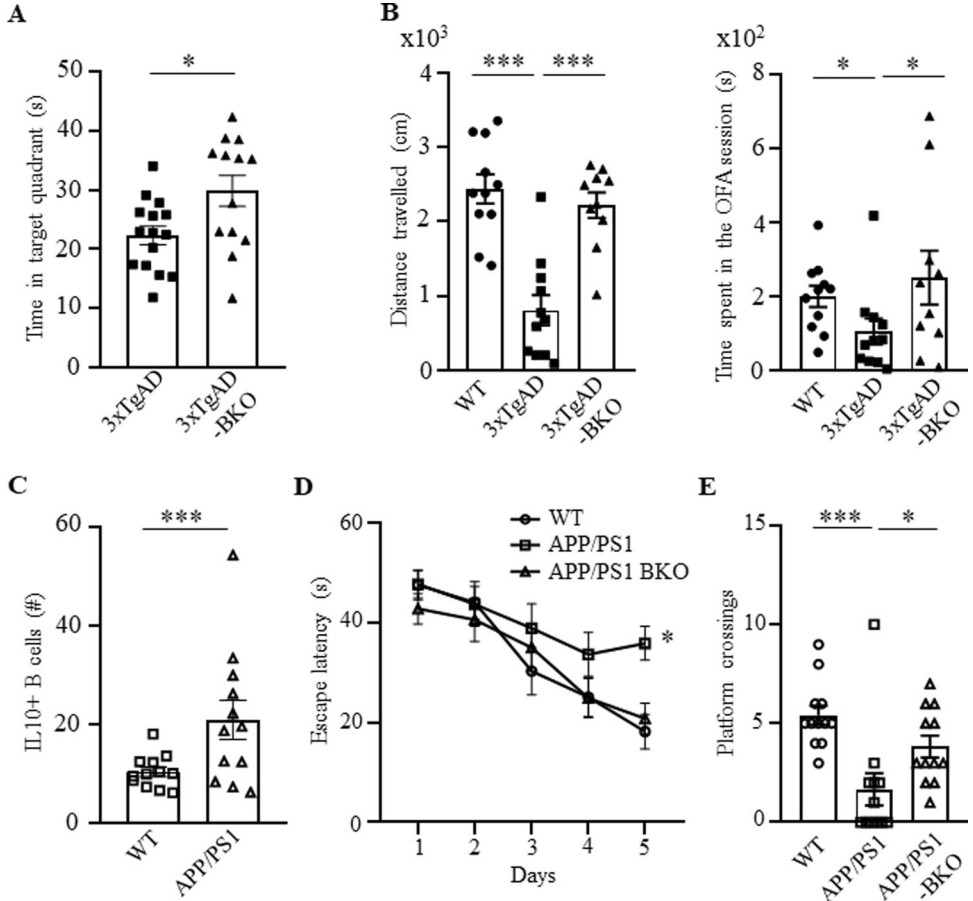

**Fig. 2 Progression of AD required B cells.** The impaired AD-associated cognitive (**A**, the MWM test) and noncognitive locomotion activity (**B**, total distance travelled, left panel, and time spent in center zone, right panel, in 30 min OFA test are shown) of 3×TgAD mice were improved in B-cell-deficient 3×TgAD-BKO mice ($n = 10-15$ female). Compared with WT littermate, APP/PS1 mice upregulated numbers of IL10$^+$ B cells in the peripheral blood (**C**). The impaired cognitive ability of APP/PS1 mice (**D**, latency to reach the escape platform; **E**, platform crossing times in a probe test in MWM test) was reversed in APP/PS1-BKO mice. Mean ± SEM is shown; each symbol is for a single mouse. **A**, **B** and **D**, **E** were independently reproduced three and two times, respectively. Gating strategy is shown in Supplementary Fig. 9A. *$p < 0.05$; ***$p < 0.001$ in unpaired $t$ test (**A**), Mann–Whitney test (**C**), one-way ANOVA (**B**, left), Kruskal–Wallis test (**B**, right and **E**) or two-way ANOVA (**D**).

APP/PS1-BKO mice (Fig. 3G and Supplementary Fig. 3H). To understand this discrepancy, we compared the function of microglia in these two models. Brain myeloid cells were isolated and stimulated with phorbol 12-myristate 13-acetate and ionomycin (PMAi, which induces microglial cytokine expression and proliferation[37]) for 4–6 h in the presence of monensin, and then surface markers and intracellular (IC) cytokines of microglia (CD45$^{Int}$CD11b$^+$) were analyzed using flow cytometry. The loss of B cells did not affect interleukin (IL)1β expression, which was markedly upregulated in both 3×TgAD and APP/PS1 (Supplementary Fig. 4A and Fig. 4A and not depicted; gating strategy is in Supplementary Fig. 9B). In contrast, we noted a marked decrease in TGFβ$^+$ and IFNγ$^+$ microglia (presumably quiescent and resting microglia[38–40]) in APP/PS1 and 3×TgAD mice (Fig. 4B, C and Supplementary Fig. 4B–D). The TGFβ$^+$ (at lesser extent IFNγ$^+$) microglial cells were normalized to the levels of WT control in APP/PS1-KO and 3×TgAD-KO mice (Fig. 4B, C and Supplementary Fig. 4B–D). Brain and hippocampal RNA microarray analyses indicated that B-cell deficiency prevented loss of TGFβ1$^+$ microglia in the hippocampi and brains of 3×TgAD mice, as its expression was upregulated (Fig. 4D) while DAM-related transcriptional signature genes, such as *Itgax, Cst7, Clec7a, Mamdc2,* and *Saa3* (ref. [4]) were downregulated in 3×TgAD-BKO mice (Supplementary Fig. 4E, Supplementary

Data 1–3, and https://www.ncbi.nlm.nih.gov/geo/query/acc.cgi?acc=GSE165111). B-cell deficiency did not affect expression of *IL1β* nor other DAM genes, such as *TNFα, Igf1,* and *Lilrb4* (Supplementary Fig. 4E and Supplementary Data 1–3). In sum, despite expression of AD-promoting transgenes, DAM accumulation, Aβ plaque deposition, and thus the disease progression required B cells in both 3×TgAD and APP/PS1 models of AD.

Next, we sought to test whether progression of EOAD can also be controlled by a therapeutic inactivation or depletion of B cells at the disease onset. To test this idea, 3×TgAD mice (60–70 weeks old, female) were intraperitoneally injected with anti-CD20/B220 antibody (which depletes B cells in the circulation) for 2 months. Control mice were treated with isotype-matched IgG. The anti-CD20/B220 Ab efficiently depleted B cells in the circulation (Supplementary Fig. 5A). Compared with control treated mice, the anti-CD20/B220 Ab-treated 3×TgAD showed a trend toward increased activity in the OFA ($p = 0.07$, 30 min, $n = 5$, Supplementary Fig. 5B). The treatment however significantly reduced the number of Aβ plaques in the subiculum of AD mice ($p < 0.05$, Fig. 4E, F), but did not affect the number of large (>50 μm$^2$) Iba1$^+$ microglial cells in the hippocampus (Supplementary Fig. 5C). To understand this discrepancy, we repeated anti-CD20/B220 Ab treatment in a different cohort of female 60–70-week-old 3×TgAD mice for 2 months ($n = 6–7$) and then performed a flow cytometry

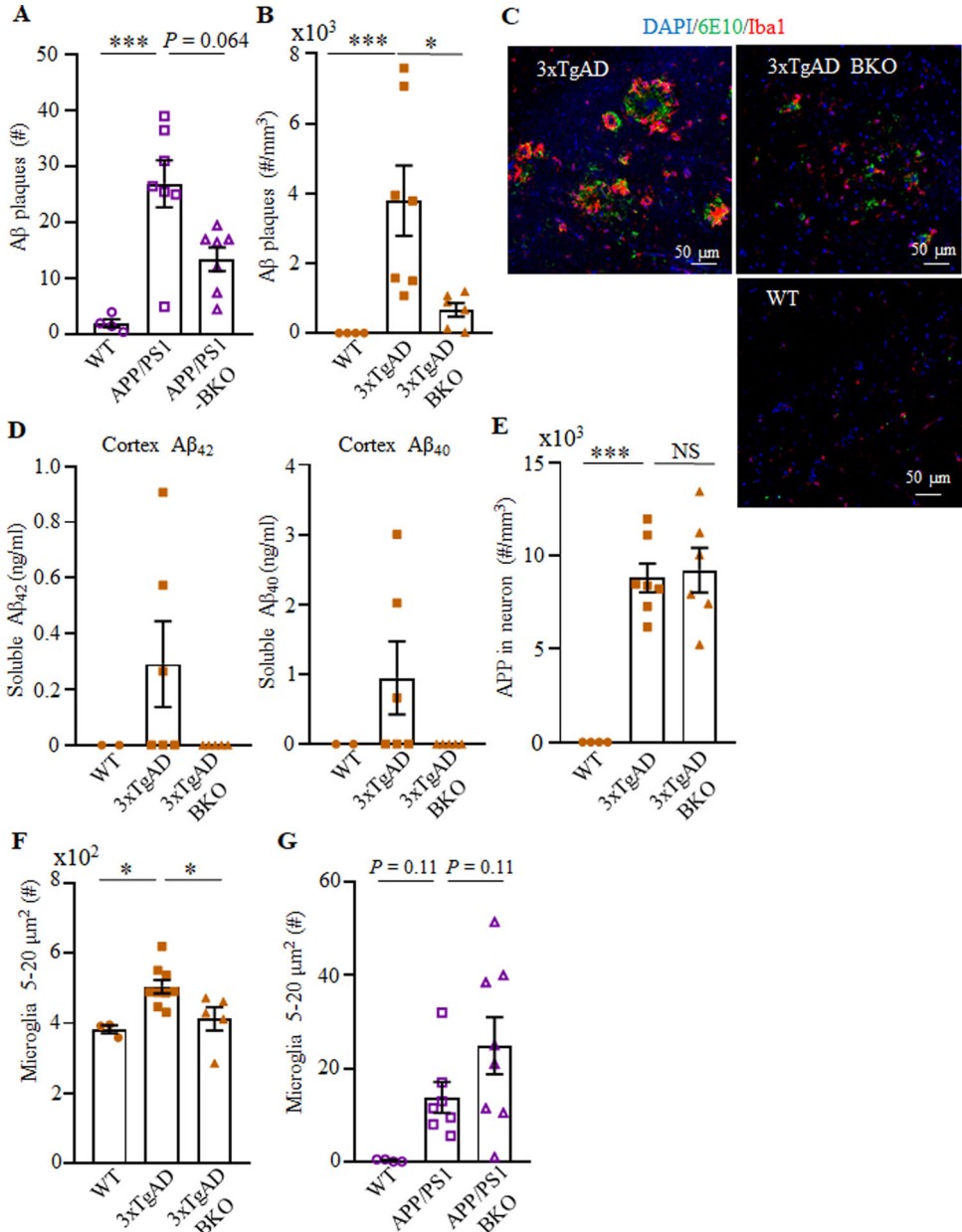

**Fig. 3 B-cell deficiency reduced the Aβ plaque burden and microglial activation in the hippocampus of AD mice. A–C** show the number of Aβ plaques in APP/PS1 (**A**) and 3×TgAD (**B**) mice ($n = 4–7$), and representative images of immune fluorescent staining of the subiculum of WT, 3×TgAD, and 3×TgAD-BKO mice (**C**, Aβ plaque (green), Iba1+ microglia (red), and DAPI (blue); scale, 50 μm). **D** The increase of soluble $Aβ_{1-40}$ and $Aβ_{1-42}$ peptides in the brain of 3×TgAD mice was reversed in 3×TgAD-BKO mice. The results of ELISA in the cortex of indicated mice ($n = 2–6$) are shown. The B-cell deficiency did not affect expression of the transgene, as both 3×TgAD and 3×TgAD-BKO highly expressed transgenic hAPP in hippocampal neurons (**E**, stained with 6E10 as in **B–C**). The B-cell deficiency significantly decreased the number (#) of large-sized microglia (5–20 μm²) in 3×TgAD ($n = 3–9$, **F**), but not in APP/PS1 mice ($n = 4–8$, **G**). Mean ± SEM is shown; each symbol is for a single mouse. *$p < 0.05$; ***$p < 0.001$; NS not significant in Kruskal–Wallis test (**A**, **B**, **G**) or one-way ANOVA (**E**, **F**).

evaluation of microglial cytokines in the perfused brains. As noted above, in 3×TgAD-BKO mice, B-cell depletion significantly reversed the decrease of TGFβ+ and IFNγ+ microglia in 3×TgAD mice to almost the levels in WT mice (both in terms of frequency and numbers, $p < 0.05$, Fig. 4C, G and Supplementary Fig. 5D, E) while not affecting IL1β+ microglia, which was comparably upregulated in aged 3×TgAD and WT mice (Supplementary Fig. 5F).

To validate these results, we used a third mouse model of EOAD, the 5×FAD mice that develop a more aggressive AD pathology by 40 weeks of age because of a large burden of Aβ plaques caused by additive effects of five familial AD mutations[41]. In this model, others have linked AD progression to a decrease in frequency of potentially beneficial B1a cells[42]. However, our cohorts of 5×FAD mice did not exhibit a decrease in B cells, including B1a and B1b cells (both in terms of number and frequency), as compared with control age-matched littermates (Supplementary Fig. 6A, B). The B cells and B1 cells in 5×FAD mice instead appeared to be activated, as they significantly upregulated expression of 4-1BBL (Supplementary Fig. 6C, D). In concordance, IC cytokine staining of splenic cells revealed marked increase of IL10, TGFβ, and IFNγ in B cells

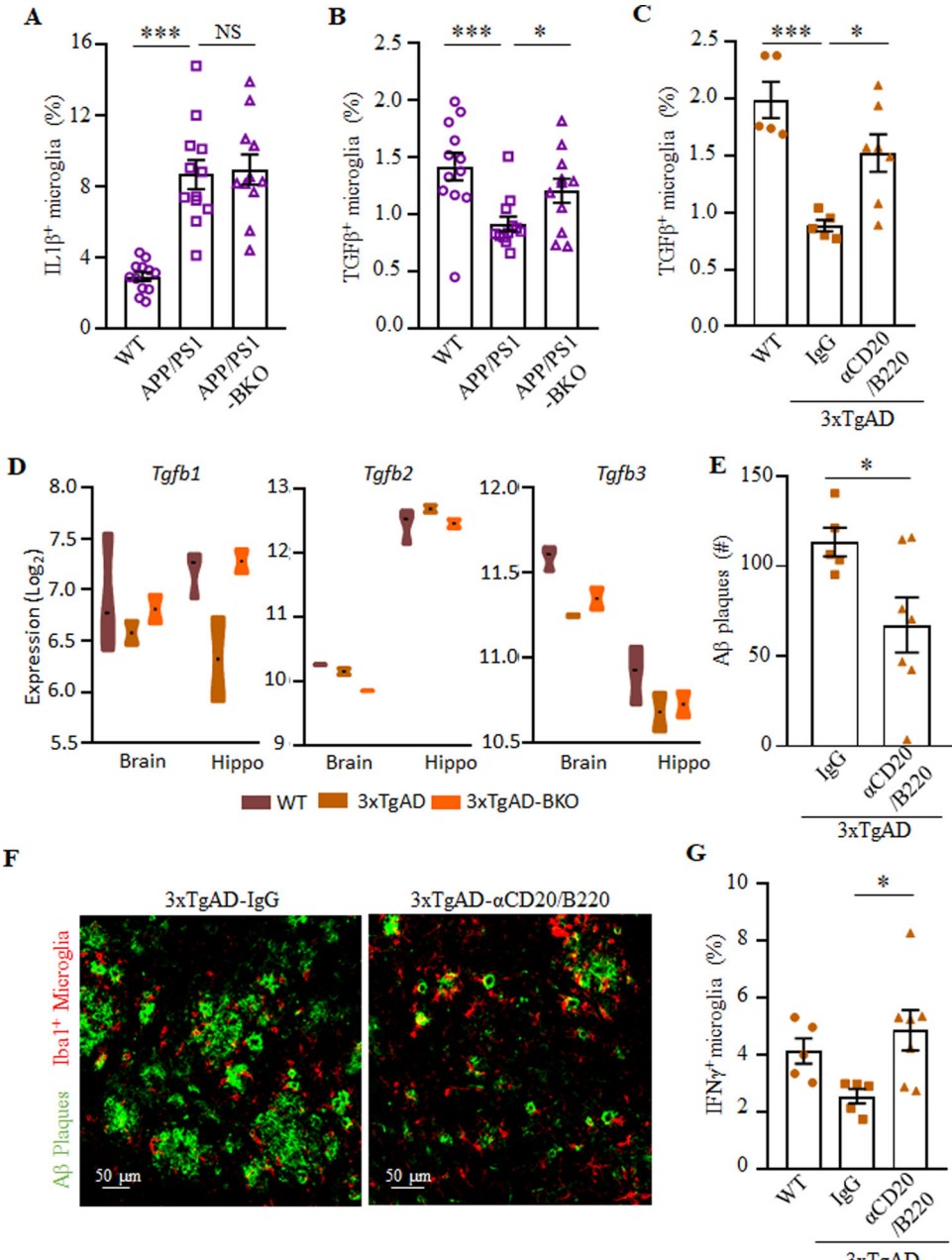

**Fig. 4 B-cell deficiency at least in part reversed the DAM phenotype.** The results of flow cytometric quantification of IL1β+ (**A**) and TGFβ+ (**B**, **C**) microglia (CD11b+CD45int) in the brains of indicated mice ($n = 11$–12) are shown. Genetic B-cell deficiency in APP/PS1 mice (APP/PS1-BKO, **B**) or transient depletion of circulating B cells (aCD20/B220, **C**) at the onset of AD (70–79 weeks of age 3×TgAD mice) reversed the AD-associated decrease of TGFβ+ microglia. mRNA microarray analyses of hippocampi and brains (without hippocampus) of 3×TgAD mice revealed that the B-cell deficiency (3×TgAD-BKO) upregulates expression of TGFβ1, but not TGFβ2 and TGFβ3 (**D**, $n = 3$). Therapeutic depletion of B cells (aCD20/B220) at the onset of AD ameliorated AD (**E**–**G**), as it markedly decreased Aβ plaques in the subiculum (quantification and representative images are shown in **E** and **F**, respectively; Aβ plaque (green) and Iba1+ microglia (red, **F**), $n = 6$–8; independently reproduced twice). B-cell depletion reversed the reduction of IFNγ+ microglia in 3×TgAD mice (**G**, $n = 5$–7). Mean ± SEM is shown; each symbol is for a single mouse. Gating strategy is shown in Supplementary Fig. 9B. *$p < 0.05$; ***$p < 0.001$ in one-way ANOVA (**A**–**C**, **G**) or unpaired $t$ test (**F**).

(Supplementary Fig. 6E). To test whether depletion of these activated B cells also ameliorates AD, 5×FAD mice (35–47 weeks old, female) were i.p. injected with anti-CD20/B220 Ab or control IgG ($n = 7$–12) for 2 months. Mice were then evaluated in the OFA for exploratory behavior and anxiety. While control IgG-treated 5×FAD mice showed reduced exploratory behavior compared with WT littermates, this effect was reversed in B-cell-depleted mice ($F(2,25) = 5.25$, $p = 0.013$; wt vs 5×FAD $p = 0.03$, 5×FAD+IgG vs 5×FAD + αCD20 $p = 0.02$, Fig. 5A). To confirm

this result, we quantified Aβ plaques and activated microglia in the hippocampus of these mice. Compared with IgG treatment, B-cell depletion in 5×FAD mice significantly reduced the number of Aβ plaques (Fig. 5B, C) and the large-sized Iba1+microglia (Fig. 5D). We repeated the 2-month B-cell depletion experiment in a different cohort of age-matched, female 5×FAD and WT mice ($n = 5$–12), and evaluated microglial cells using flow cytometry in perfused with saline brains. Similar to 3×TgAD and APP/PS1 mice, TGFβ+ and IL10+ microglia were significantly

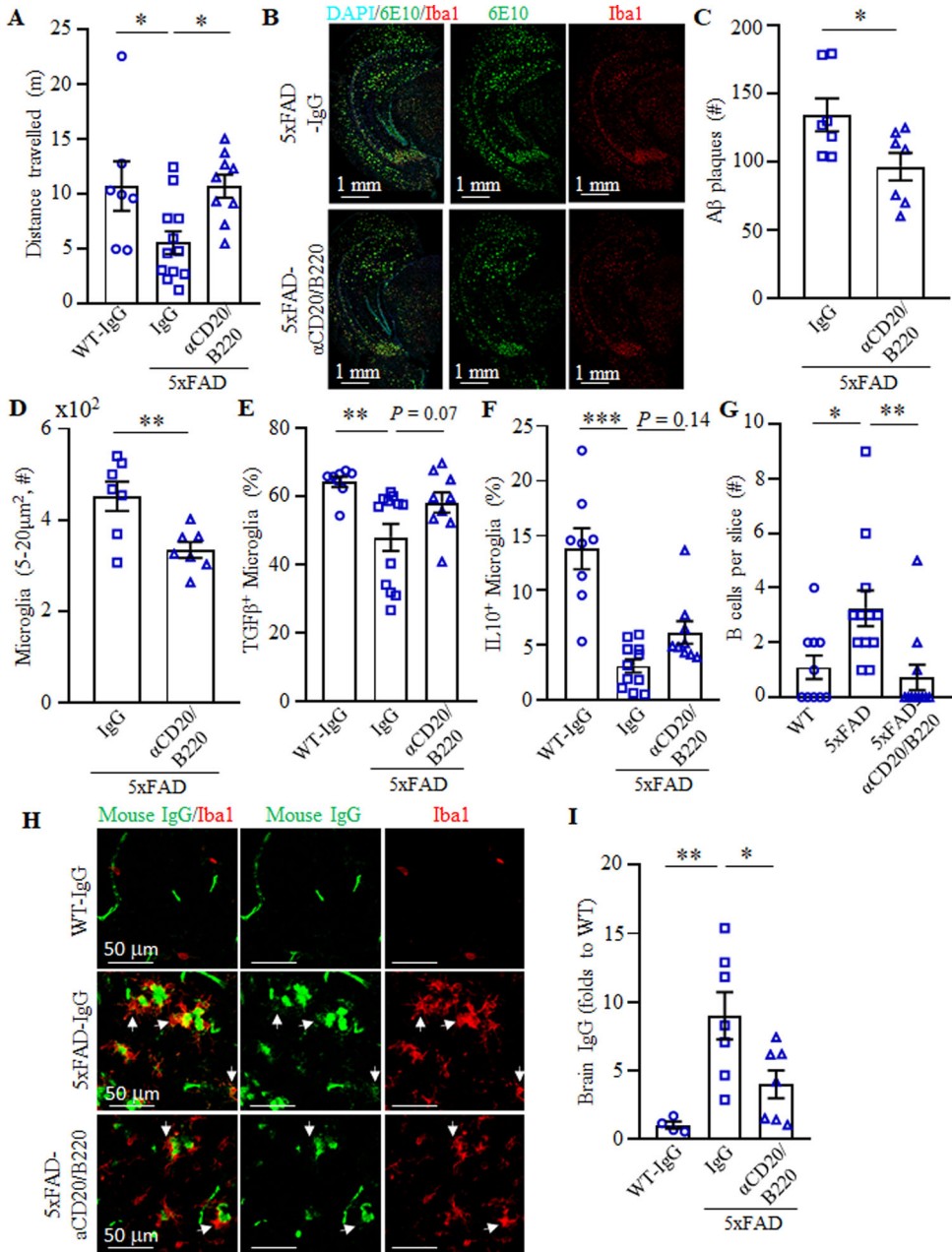

**Fig. 5 Therapeutic depletion of B cells improves AD symptoms in 5×FAD mice.** OFA test revealed that B-cell depletion improves the retarded locomotion of 5×FAD mice (**A**, total distance travelled in first 5 min). B-cell depletion reduced Aβ plaque burden in the hippocampus (**B**, **C**). Representative immune fluorescent staining images for Aβ plaques (green), Iba1+ microglia (red), and DAPI (cyan) are shown in **B**. In **C** and **D**, the results of quantification of Aβ plaques and microglia are shown (**D**, large size, 5–20 μm$^2$), respectively. **A–D** Results were reproduced twice, $n = 7$–12. Flow cytometric evaluation of brain microglia (CD11b+CD45$^{int}$) revealed that B-cell depletion increases the frequency (%) of TGFβ$^+$ (**E**) and IL10$^+$ (**F**) microglia in 5×FAD mice ($n = 10$–12). As shown with immune fluorescent staining, B220$^+$ B cells were markedly increased in the brain parenchyma of 5×FAD compared with WT controls, which was lost after transient B-cell depletion (**G**, $n = 9$–12). The brains of 5×FAD mice contained high levels of IgG as compared with WT controls, which was also reversed by transient depletion of B cells (**H** and **I**). In **H**, IgG is green and Iba1$^+$ microglia is red. Brain IgG quantification is in **I** ($n = 5$–7). Mean ± SEM is shown; each symbol is for a single mouse. Gating strategy is shown in Supplementary Fig. 9B. *$p < 0.05$; **$p < 0.01$; ***$p < 0.001$ in one-way ANOVA (**A**, **E**, **F**, **I**), Kruskal–Wallis test (**G**), or unpaired $t$ test (**C**, **D**).

decreased in 5×FAD mouse brains as compared with that of WT mice ($p < 0.001$, Fig. 5E, F). B-cell depletion reversed the decrease of TGFβ$^+$ and IL10$^+$ microglia in 5×FAD mice (Fig. 5E, F). As in other models, the treatment did not decrease hippocampal IL1β$^+$ microglia (Supplementary Fig. 7A, B), presumably to support clearance of Aβ plaques[43]. Collectively, we concluded that depletion of B cells can retard progression of AD even if applied at the onset of the disease.

We then wondered whether B cells promote AD by infiltrating the brain parenchyma. Since the low numbers of B cells (<1% of CD45$^+$ cells, not depicted) detected in flowcytometric analyses cannot discriminate between cells residing inside or outside of the parenchyma, we performed immune fluorescent staining of per-fused and cryopreserved brain sections of mice. Unlike control age-matched WT mouse brains, we clearly detected a significant increase of B cells in the parenchyma of frontal cortex and

hippocampus of 5×FAD mice (Fig. 5G and Supplementary Fig. 7C–E). Importantly, this increase of B cells was almost completely lost in B-cell-depleted 5×FAD mice (Fig. 5G and Supplementary Fig. 7D, E), implying that the cells originated in the circulation. We also stained the brain sections for the presence of IgG. While WT mouse brains were almost devoid of immunoglobulin (Fig. 5H, I), readily detectable and numerous IgG foci were present in the hippocampus parenchyma of 5×FAD mice often colocalized with microglia and Aβ plaques (Fig. 5H, I). This brain IgG increase was lost in B-cell-depleted 5×FAD mice (Fig. 5H, I). However, the depletion did not affect levels of total IgG nor minuscule amounts of Aβ-specific antibody in the circulation (Supplementary Fig. 8A–C), implying that B cells in the brain parenchyma produced IgG and presumably promoted AD.

## Discussions

Taken together, we provide counterintuitive evidence for a "dark" side of B cells—they exacerbate manifestation of AD-like symptoms in addition to producing potentially beneficial Aβ plaque-reducing immunoglobulins[17–21] and expressing AD-ameliorating cytokines[42]. Although the exacerbation in Rag-deficient APP and 5×FAD mice is linked to the loss of protective B cells and T cells[16,21], our data revealed that the genetic loss of B cells alone or their transient depletion at the onset of AD improves the disease symptoms of three different mouse models. Unlike a recent report that linked AD progression to the reduction of anti-inflammatory B1a cells in 5×FAD mice[42], the numbers of B1a and B1b cells in PB, spleen, and cLN were either unaffected (in 5×FAD mice even when followed for 4, 7, and 12 months) or upregulated (in 3×TgAD and APP/PS1 mice). However, regardless of their numbers, we recently reported that the function of B1a cells is not static and is rather controlled by the inflammatory milieu. In the aged hosts, B1a cells lose their anti-inflammatory activity and acquire pathogenic functions, such as becoming 4-1BBL+ B1a cells (termed 4BL cells) that induce cytolytic granzyme-B+ CD8+ T cells and promote insulin resistance[23,44]. In concordance, B1 cells (as well B2 cells, in some models) in AD mice also appeared to acquire an inflamed phenotype, as they upregulated expression of cytokines, such as IFNγ, IL6, TNFα, and IL10, and/or upregulated TGFβ and 4-1BBL. Although age-associated B cells (CD21− CD23− CD19+) also accumulate in aging, we did not detect their involvement in our three types of mice with AD.

Consistent with a recent RNA-seq report that revealed presence of mature B cells in the brains of AD mice[4], our data indicate that AD increases B cells in the brain, and their IgG in the cortex and hippocampus parenchyma, which was often colocalized with Aβ plaques and activated microglia. As in multiple sclerosis and cognitive dysfunction following stroke[45–47], B cells in the brain presumably produce immunoglobulins and proinflammatory factors exacerbating AD-promoting neuroinflammation. First, transfer of IgG from the circulation into the brain is an inefficient process, as only 0.0017% of intravenously injected immunoglobulin reaches the hippocampus of WT and AD mice[48]. Second, consistent with presumed ability of the intra-blood-brain-barrier sites of AD patients to synthesize immunoglobulin[49], transient depletion of B cells in 5×FAD mice significantly decreased brain B cells and IgG without affecting Aβ-specific and nonspecific antibody levels in sera. Although immunoglobulin is thought to activate and promote microglial uptake of Aβ plaques[21], the Aβ–IgG complex in CSF is thought to negatively affect the cognitive status of AD patients[50]. Our data also indicate that the loss of B cells, thus IgG, in the brain significantly retards the development of AD. Although the mechanism of this process is a topic of a different study, we think

that brain IgG (or its immune complex) alone or in concert with B-cell cytokines exacerbates neuroinflammation in AD. To do this, the brain IgG presumably targets chronically stressed DAMs (and other brain cells)[4] through their upregulated Fc-receptors and complement[51–53], as recently shown for myelin–IgG immune complexes from the brain of people with multiple sclerosis, which break immune tolerance of human microglia to microbial stimuli and cause harmful neuroinflammation via FcγRI and FcγRIIa[54]. This in turn leads to downregulation of TGFβ+ microglia, i.e., reduction in survival of resting M0 microglia and Aβ plaque clearance[3,40,55]. Similar decrease in expression of TGFβ in DAM and increase in signals of mature B cells is also noted in recent brain RNA-seq results of AD mice[4]. Conversely, the loss of B cells increased TGFβ+ microglia as well downregulated expression of Trem2, Clec7a, and Itgax in the hippocampus, i.e., replacement of DAM with microglia that eliminate Aβ oligomers and other neurotoxic debris[8–10], reduced Aβ plaques and improved behavioral impairments of our AD mice. It is tempting to speculate that B cells similarly participate in EOAD in humans, as the severity of the disease in humans with mild AD is correlated with accumulation of double negative memory CCR6+ B cells in the circulation[56]. Moreover, elderly humans accumulate pathogenic 4-1BBL+TNFα+ B cells that induce antigen-specific CD8+ T cells and promote insulin resistance[22,23], two events linked to AD[57,58]. We therefore propose that the inactivation of B cells can also benefit humans with AD, as therapeutic B-cell removal even at the onset of the disease reversed manifestation of AD in mice.

## Methods

**Mice.** The animal protocols were approved and permission was granted to perform animal experiments by the ACUC committee of the National Institute on Aging (ASP 321-LMBI-2022) under the Guide for the Care and Use of Laboratory Animals (NIH Publication No. 86-23, 1985). AD transgenic mice (5–80 weeks of age, females) were bred, aged, and housed in the same, specific pathogen-free environment at the National Institute on Aging (NIA). Female C57BL/6j mice (Stock # 000664) were purchased from Jackson Laboratory (Bar Harbor, ME) and congenic 3×TgAD mice (triple transgenic with three human genes associated with familial AD, B6;129-Psen1tm1Mpm Tg(APPSwe,tauP301L)1 Lfa/Mmjax)[24,25], APP/PS1 mice (B6.Cg-Tg(APPswe,PSEN1DE9) 85Dbo/J)[16], 5×FAD mice expressing mutant human APP and PSEN1 genes (B6.Cg-Tg;APPSwFILon,PSEN1*M146L*L286V)[21] and JHT mice (B6.129P2-Igh-Jtm1Cgn/J), which do not develop functional B cells in the circulation due to the immunoglobulin JH locus deletion[26], were bred and maintained at NIA, Baltimore, MD. Transient B-cell depletion was performed as previously reported[22,59], such as mice were injected intraperitoneally (i.p., three to six times for 2–3 months) with anti-CD20 Ab (clone 5D2, Genetech, 100 μg/mouse) and anti-B220 Ab (150 μg/mouse, TIB-146, BioXCell).

**Tissues and blood processing.** Single-cell suspension of spleen and cLN was prepared using a 70 μm cell strainer (BD Falcon, Bedford, MA). Blood was collected in tube with 2 mg/ml of Na-heparin (Sigma). Spleen and blood cell suspensions were treated with ACK buffer to remove red blood cells. The brains were dissociated with Adult Brain Dissociation kit for mouse and rat (MiltenyiBiotec, Auburn, CA) using GentleMACS™ Dissociator (MiltenyiBiotec), following the manufacturer's instruction.

**Flow cytometry (FACS).** Antibodies (Ab, see Supplementary Data 4) to mouse CD19, CD5, CD11b, CD43, 4-1BBL, IFNγ, IL1β, TGFβ, IL10, IL6, CD11b, and CD45 and their isotype-matched control Ab were purchased from Biolegend, eBioscience, BD Bioscience, and R&D Systems, unless specified. For IC cytokine staining, cells were stimulated with 50 ng/ml PMA (Tocris Bioscience) and 500 ng/ml ionomycin (Tocris Bioscience) for 1–2 h, followed by Golgi stop for 3–4 h using 10 μmol/l of Monensin or Brefeldin A (eBioscience); and then stained following manufacturer's instruction for IC fixation and permeabilization (eBioscience). Concentration of antibody used in FACS staining was 1 μg per 10^6 cells. Data were analyzed on FACS Canto II (BD) or CytoFLEX (Beckman Coulter, Inc.) using FlowJo software (Tree Star, Inc.) or CytExpert software (Beckman Coulter, Inc.).

**Brain immune fluorescent staining.** Mice were perfused with PBS for 20–30 min after euthanasia with CO₂, and the brains were removed and washed with PBS, and then half brain was fixed by 4% PFA in PBS. After 24-h fixation, the brain was transferred to PBS buffer containing 30% sucrose in 15 ml tubes for ~2 days until the brain sank to the bottom. Next, the brains were embedded in OCT compound, frozen on dry ice and stored in −80 °C before cryosection. A total of 30-μm thick coronal

sections containing hippocampus were collected with 240 μm interval to get eight sets and two to six slices of each mouse were stained for each immunofluorescence staining. For immunofluorescence staining, we adopted free-floating staining method: after two washes with PBS, the brain slices were incubated in 0.3 M glycine buffer for 30 min at room temperature (RT), and then blocked and permeabilized with IF buffer (2% donkey serum, 2% BSA, and 0.1% Triton X-100 in PBS) for 30–60 min at RT. Brain slices were incubated with designated Abs or their isotype control immunoglobulins purchased from Abcam, overnight at 4 °C followed by 15–60-min incubations at RT. After three washes with PBS, brain slices were incubated with fluorochrome-conjugated secondary Abs from Abcam (Donkey anti-mouse IgG H&L-Alexa Fluor 488; Donkey anti-rabbit IgG H&L-Alexa Fluor 568; Donkey anti-rat IgG H&L-Alexa Fluor 568; Donkey anti-goat IgG H&L-Alexa Fluor 647;) at RT for 2 h. After three washes with PBS, slides were mounted with ProLong™ Gold/Diamond Antifade Mountant with DAPI (Invitrogen). For IgG staining, brain slices were incubated with Iba1 antibody for 2 h at RT followed by wash with PBS, and then incubated with Alexa Fluor 488 labeled F(ab′)2-Goat anti-Mouse IgG (H + L) Antibody (cat # A-11017) and Donkey anti-rabbit IgG H&L-Alexa Fluor 568. The information of the first antibodies used is as below: purified anti-β-Amyloid, 1-16 Antibody (clone 6E10, cat # 803002), Recombinant Anti-Iba1 antibody (cat # ab178846), Mouse IL1 beta /IL-1F2 Antibody (cat # AF-401-NA), Anti-CD45R (B220) antibody (cat # ab64100), ZO-1 Polyclonal Antibody (cat # 40-2300), and Mouse Laminin alpha 4 Antibody (cat # AF3837).

Images were acquired with a Zeiss LSM 710 confocal microscope equipped with a ×20/0.8 and a ×20 Plan-Apochromat dry objectives (Carl Zeiss). For Aβ plaque and microglia quantification, subiculum region for 3×TgAD model, or dentate gyrus region for APP/PS1 model, or whole hippocampus region for 5×FAD model were respectively identified in the coronal brain slices and imaged. For brain IL1β or IgG quantification, subiculum region was imaged. For brain B220$^+$ B-cell quantification, the whole brain was observed and images for B220$^+$ B cells were captured. The quantification of all images was performed with ImageJ software. For Aβ and microglia quantification, the image of up to six slices per mouse were acquired and quantified in the subiculum region (3×TgAD model), DG region (APP/PS1 model), or whole hippocampus (5×FAD model). Briefly, the single channel image was converted into 8-bit image and proper threshold was chosen and fixed for all the images in the same experiment, then "analyze particles" function was used for quantified the number or area of 6E10 plaque and Iba1$^+$ microglia. The representative image for microglia quantification was shown in Supplementary Fig. 3G. For B220$^+$ B-cell quantification, the whole brain region was carefully observed and the images of B220$^+$ B cells were acquired, and the location of B cells was distinguished by ZO-1 or laminin a4 staining or bright field.

**Measurement of soluble Aβ$_{40/42}$ peptides.** Cerebral cortex and hippocampus were separated from PBS-perfused mouse brains, and stored at −80 °C. Soluble and insoluble protein fractions were purified using a modification of previously published protocol for Aβ peptides[60]. Briefly, tissues were mechanically homogenized in TBS-Triton 1% (120 mM NaCl, 50 mM Tris, pH 8.0, 150 mg/ml (tissue/buffer)), including protease inhibitor cocktail (1:100, P2714, Sigma, St. Louis, MO), then incubated on ice for 30 min followed by centrifugation for 120 min at 17,000 × g at 4 °C. Supernatant, containing TBS-T-soluble fraction of mouse Aβ1–40 and Aβ1–42 was removed and stored at −20 °C. The remaining pellet, containing insoluble Aβ was resuspended in 70% formic acid and incubated on ice for 30 min, followed by centrifugation at the same conditions as mentioned above. Formic acid-soluble supernatant was separated and neutralized using 1 M Tris (pH = 11, 20-time the volume of the formic acid) and stored at −20 °C. Total protein concentration was determined using the BCA method (cat # 23225, Thermo Fisher Scientific, Waltham, MA). TBS-S and formic acid soluble levels Aβ$_{1–40}$ and Aβ$_{1–42}$ in the cortex were measured using sandwich-ELISA protocol in 96-well polystyrene microplates (655061, Greinerbio-one, Monroe, NC) were covered with 50 μl of anti-rabbit-N-terminus Aβ$_{1–14}$ (ab2539, Abcam, Cambridge, UK) at a concentration of 5 μg/ml in carbonate–bicarbonate buffer (pH = 9.6) and incubated overnight at 4 °C. Plates were washed four times in PBS-T solution (0.1% Triton X in PBS) and blocked with 2% BSA solution in PBS. A total of 50 μl of tissue homogenate were applied to each well, and incubated for 60 min at RT. Plates were then washed five times in PBS-T, and the following detection antibodies were added: Anti-Aβ$_{1–40}$ antibody (ab20068, Abcam, Cambridge, UK) diluted at 1:500 or anti Aβ$_{1–42}$ antibody (05-831, Millipore, Billerica, MA) at 1:2500, and incubated for 60 min at RT. Next, plates were washed five times in PBS-T and secondary goat anti-mouse IgG HRP-conjugated antibody was added (115-035-003, Peroxidase AffiniPure Goat Anti-Mouse, Jackson immunoresearch, PA) at a dilution of 1:5000. Plates were washed five times with PBS-T and 50 μl of 3, 3′, 5, 5′-tetramethylbenzidine substrate (00-4201-56, Affimetrix eBioscience, San Diego, CA) was added. The color reaction was allowed to develop for 3 min and was stopped by adding 50 μl of 2 M H$_2$SO$_4$. Optical density was measured at 450 nm using a spectrophotometer. Standard curve was carried out using known concentrations of recombinant Aβ$_{1–40}$ and Aβ$_{1–42}$.

**Behavioral tests.** For the MWM test, we used a 140-cm-diameter tub containing a 15 × 15 cm square platform submerged 1 cm below the water surface. Water was maintained at 21 ± 1 °C and colored with white paint. Mice underwent four trials per day, released from each of four compass points (N, S, W, E) in a randomized order. If a mouse did not find the hidden platform within 60 s, it was gently guided to the platform. All mice remained on the platform for 10 s at the end of each trial,

were towel tried, and returned to their home cage. After 5 days of training mice were subjected to a 60-s probe trial that took place 24 h after the last training trial. The platform was removed and mice were placed on the opposite wall at the point furthest from the former platform location. The day after the probe trial, mice were tested for four trials in their ability to swim to a visible platform. Swimming was analyzed with ANY-Maze software (Stoelting). For the open field test, mice were placed in a 34 × 34 × 25 cm white acrylic chamber and recorded by overhead camera for 60 min. Because the first few minutes in this test are maximally sensitive to AD transgenic mice[30], we focused analysis on the initial 5 minutes. Tracks were analyzed with ANY-Maze software. Analysis in all behavior experiments was performed using ANY-Maze software. For the gait analysis, mice were tested on a fixed-speed treadmill apparatus (DigiGait; Mouse Specifics). Mice were habituated to the apparatus for 1 min, and then given a 1-min run at 5 cm/s. Following a 1-min rest, the treadmill speed was increased to 15 cm/s. Video was collected at high speed from a ventrally placed camera, and 3–5 s of representative gait video was selected by an experienced user for automated analysis.

**Microarray analysis.** From PBS-perfused brains or hippocampi of mice, total RNA was isolated with RNeasy Plus micro kit (QIAGEN) according to manufacturer's instruction. Microarray analysis was performed using the Illumina and Agilent platforms, then we quantile-normalized the gene expression profiles[61], log2-transformed the raw probe intensity values and estimated the gene expression using the probe with the highest average expression[62]. Microarray data are submitted at https://www.ncbi.nlm.nih.gov/geo/query/acc.cgi?acc=GSE165111.

**Immunoglobulin ELISA.** Blood was collected in BD Microtainer® Tubes and serum was separated by centrifugation following the manufacturer's instruction. For Aβ-specific immunoglobulin ELISA, sera (diluted in 0.05% Tween20/2% BSA/PBS) were incubated in 96-well plates coated with 4–8 μg/ml Aβ$_{1–42}$ peptide (AnaSpec, cat # AS-24224) and quantified using HRP-conjugated goat anti-mouse IgA, IgM, or IgG antibodies. For serum immunoglobulin ELISA, Ig Isotyping Mouse Uncoated ELISA Kit (Invitrogen, Cat No. # 88-50630-88) was used, following the manufacturer's instruction.

**Statistical analysis.** All statistical analyses were performed with GraphPad Prism (Prism 8; Graph Pad Software, Inc.). Normality tests were conducted to decide whether the data are from normally distributed population. Unpaired t test or one-way ANOVA, for two groups or three groups, respectively, was used if the data are considered to be normally distributed, otherwise Mann–Whitney test or Kruskal–Wallis test was used for two groups or three groups, respectively. Due to changes in variance across days of invisible platform testing in MWM, as is typical in this task[63], each day was individually analyzed using one-way ANOVA. Bonferroni post hoc tests were conducted when appropriate. The results were presented as the mean with each individual data point or in bar graph ± SEM. A p value <0.05 was considered significant (*$p < 0.05$, **$p < 0.01$, and ***$p < 0.001$).

**Reporting summary.** Further information on research design is available in the Nature Research Reporting Summary linked to this article.

## Data availability

All data associated with this study can be found in the paper or supplementary materials. The microarray data are deposited at https://www.ncbi.nlm.nih.gov/geo/query/acc.cgi?acc=GSE165111. Source data are provided with this paper.

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

## Acknowledgements

We are grateful to Dr. A. C. Chan (Genentech, Inc.) for providing anti-CD20 Ab; Drs. K. Becker, E. Lehrmann, and Y. Zhang (NIA) for microarray assay and bioinformatics analysis; and Mrs. A. Lustig (NIA) and Dr. Chandamany Arya (Lilly) for proofreading. This research was supported by the Intramural Research Program of the National Institute on Aging, NIH, and by grant NIH R01 AG054712-01A1.

## Author contributions

K.K., X.W., E.Ra., and M.B. performed the research and collected and analyzed data; T.I., M.D., and R.M. performed experiments; F.G. provided bioinformatics support, E.Ro. and E.O. provided critical interpretation; and A.B. wrote the manuscript and conceived, designed, and supervised the study.

## Funding

## Competing interests

The authors declare no competing interests.
