## [Peer Review File · Nature Communications]

Reviewers' comments:

Reviewer #1, expert on Alzheimer's disease (Remarks to the Author):

This is an interesting manuscript that shows that depleting B cells in 3 mouse models of AD improves performance in the Morris water maze and reduces A β levels and disease associated microglia. The premise of the manuscript is interesting and suggests a potential role for B cells in the pathogenesis of AD. The use of 3 different AD models makes the findings more convincing. The manuscript however is difficult to read and requires some heavy editing and has several typos. More importantly the manuscript is missing some additional data that would further strengthen the findings and improve its impact as detailed below.

1-In figure 1, since the manuscript does not really show any mechanism that explains the role of B cells in AD, it should at least include better characterization of the dynamics of B cell involvement. So additional time points for the numbers of the various B cells should be included. It would also be important to show these dynamics and numbers in the PS1APP model since the behavioral testing shown in figures 1F and 1G were done in this model. It would also be important to show the presence of B cells in the brains of these mice, either by histology or by flow cytometry.

2-In figure 2, representative images of the whole brain from 3xTg mice and PS1APP mice should be shown as well as quantification of plaques and soluble A β . Showing bits and pieces of data from different mice is not adequate.

In addition, it is not clear why is the focus only on A β . 3xTg mice also exhibit increased Tau phosphorylation. What happens to Tau as a result of B cell depletion in these mice? This should be easy to test.

3-In figure 3, what happens to A β plaque numbers in areas other than the subiculum? The data should be shown.

4-In figure 4, again data from the whole brain should be shown as well as representative images.

5-In various figures, when the authors refer to large microglia it would be helpful to show representative images so that the reader can have an idea how these microglia look and additional parameters to confirm these are microglia (and not monocytes for example) should be used.

6-A general comment is that deposition of A β and the presence of DAMs are not "symptoms" of AD as the manuscript repeatedly states. They are pathologic manifestations of the disease. This should be corrected throughout the manuscript including abstract.

Reviewer #2, expert on B cells in autoimmunity (Remarks to the Author):

This group used 3 mouse models of Alzheimer's disease (AD) to show that B cell deficiency alone prevents AD symptoms. This work suggests that expression of AD transgenes alone is not sufficient to drive disease, it requires B cells. This result suggested that targeting B cells should be considered for the treatment of AD. This was done therapeutically in the mouse AD models, using B cell-depleting agents and it showed efficacy in preventing AD. This work comes at a time when a number of other impactful articles are suggesting a strong involvement of the immune system in AD.

AD affects older patients and is caused by the impaired clearance of toxic protein aggregates from the brain parenchyma, such as amyloid-beta (A-beta) peptides of aberrantly cleaved Amyloid Precursor Protein (APP). Microglia which normally control A-beta toxicity via production of TGF-beta, are over-activated in AD, triggering inflammation. Disease progression is also function of peripheral inflammation, an aspect that is more common in older patients. The role of the immune system in AD was initially shown using RAG mice but the work never distinguished the respective contribution of B and T cells. Previous reports have suggested that the loss of non-specific Ig, which activate microglial phagocytosis of A-beta, is thought to exacerbate AD in some animal models, suggesting a pathogenic role of B cells.

This group used 3xTgAD mice with 3 human genes associated with familial AD. Female were used due to their more pronounced AD.

Fig 1: Shows increased numbers of B cells that are A4-1BBL+ and IFN-gamma+ in 3xTgAD mice

resembling aging B cells in elderly patients but there is no phenotyping of these B cells as B-1a B cells have been shown to also express 4-1BBL. Not clear whether these B cells might be ABC B cells seen in ageing animals (not enough markers were used to distinguish/identify potential B cell subsets.).

Cognitive functions were restored in 3xTgAD mice lacking B cells and 3xTgAD mice lacking B cells had reduced Hippocampal A-beta plaque burden of AD and reduced microglia number in lesions (Fig. 2). This was validated using a separate model of AD APP/PS1/BKO. B cell deficiency up-regulates TGF-beta without affecting IL-1-beta production, shown using B cell-KO AD mice or following depletion of B cells (Fig. 3). B cell depletion also reduced AD symptoms in a third model of AD (Fig. 4). The data suggest that B cell depletion improves IL-10 expression by microglia but control microglia from WT mice treated with control IgG express a lot more IL-10, and IL-10 production from microglia isolated from AD animals in which B cells have been depleted remains very impaired albeit slightly better than that of AD mice with B cells.

The issue with this work is that there is no mechanism. Previous work has shown a role for antibodies. This was not tested here. For instance, what happens when purified Ig from AD mice are injected in AD-BKO mice? I think this is important considering that no B cells were detected in lesions, and a role for antibodies in AD has already been shown. The main question is about the specificity of the B cells expanded in AD. There is little attempt to characterize these B cells and explore the pathogenic mechanism they trigger in disease.

In conclusion, as it stands this work may not be novel and just confirm the already described role of antibodies in AD.

Reviewer #3, expert on B cells immunopathology (Remarks to the Author):

I have reviewed the manuscript entitled "Therapeutic B-cell Depletion Reverses Progression of Alzheimer's Disease" by Kim et al.

Using three different mouse AD models, the authors report that B-cell deficiency alone retards AD symptoms, such as A β plaque burden and behavioral and memory deficits by eliminating the disease-associated microglia and restoring TGF β + microglia and further claim that therapeutic depletion of B cells even at the onset of the disease was sufficient to reverse AD progression.

To answer this question, the authors generated murine crosses between JHT mice harboring the immunoglobulin JH locus deletion with three distinct murine models of AD (3xTgAD, APP/PS1 and 5xFAD). Descriptive effects on behavioral and pathology outcomes were assessed and differences, when significant, were deemed to be due to Ig+ B-cell agenesis.

The authors concluded that despite expression of AD-promoting transgenes, the disease progression requires B cells and, thus, their therapeutic depletion can also be beneficial. To test this possibility, the authors depleted B cells at the onset of the disease in 60-70- weeks old, female 3xTgAD mice by intraperitoneally injecting anti-CD20/B220 antibody (Ab) for 2 months. Appropriate controls were treated with isotype-matched IgG. Compared with controls, B-cell-depleted mice only slightly improved behavioral performance in OFT and did not affect the numbers of hippocampal larger-sized Iba1+ microglia but significantly decreased A β plaques.

This report builds upon the authors' previous work highlighting B-cell functionalities, distinct from ability to make Ig, in participating in human pathology. The concept is meritorious especially since CD19/CD20 B-cell depletion is actionable in humans with current FDA-approved drugs in common clinical practice.

Major issues:

The authors highlight the potential neuroprotective benefit of vaccine-induced or naturally occurring antibodies to A β plaques and amyloidogenic peptides in peripheral blood and cerebrospinal fluid of people and mice with AD and further inform the readership that even loss of

non-specific immunoglobulins, which activate microglial phagocytosis of A β , is thought to exacerbate AD (ref #20). Despite this relevant preamble, the humoral immune characterization of BKO crossed AD mice is completely lacking. The authors appropriately state that non-specific Igs, Igs with anti-amyloid specificity and B-cells themselves (independently of their ability to make Igs) can affect AD biology. In the original reporting of JHT mice harboring the immunoglobulin JH locus deletion (reference 25), it is clearly stated that these mice lack IgG and IgM and are essentially agammaglobulinemic. Therefore, all the BKO crosses described in this report likely lack Igs as well as B-cells. This experimental approach makes it impossible to separate the effect of absence of Igs from the non-Ig functionalities of B-cells [eg: TNF α production and the like advanced by the authors].

The authors further state "Although the molecular mechanism of this process remains unknown and is a topic of a different study, we think that AD induces pathogenic B cells to remotely exacerbate the A β -induced DAM phenotype by inhibiting microglial expression of TGF β 1."

This explicit admission highlights the major flaw of this report, namely the descriptive nature of results which do not interrogate two non-overlapping immune functionalities: (i) non-specific Igs and A β -specific Igs from (ii) B-cell function distinct from Ig production. In the genetic crosses used to generate BKOxAD mice, there is no determination whether these mice are hypo or agammaglobulinemic. Assuming these mice lack Igs, is benefit from absence of Igs or absence of non-Ig B-cell functionality?

This shortcoming is vexing considering the data contained in Reference #20 which clearly suggested a "protective" role of B-cells in AD pathology in mice [which runs contrary to the hypothesis here pursued]..

In another series of experiments mice are treated with B-cell depleting monoclonal antibody, yet the assessment of B-cell depletion on Ig content by this method is not provided. What is Ig content in these mice post treatment? Presumably mature plasma cells in marrow and spleen are unaffected. Is there any measurable anti-amyloid activity remaining in these mice?

Most substantially, the data presented is descriptive with substantial gaps in mechanism of action of B-cell depletion and we are told this type of work will be performed latterly. This would need to be part of this corpus of work.

Minor issue:

The treatment protocol of mice with anti-B220 antibody for B-cell depletion is not provided [eg: dosing, frequency] nor is there a reference validating/describing use of this approach

Authors' point-by-point response to reviewers' comments

We thank the reviewers for their time and helpful suggestions. Below is our response to your comments, which whenever applicable, we addressed by performing additional experiments. Our manuscript has been substantially revised following your suggestions. Besides improving its presentation style and clarity, we included additional new results to emphasize the novelty of our findings, such as:

1. B cells are required for progression of AD of every mouse model we tested (the three most widely utilized types of transgenic AD mice)
2. We link AD to increased presence of B cells and their IgG in the brain parenchyma, including hippocampus
3. Transient depletion of circulating B cells is sufficient to eliminate accumulation of B cells and IgG in the brain and to retard the development of AD symptoms, suggesting that B cells should be considered as therapeutic targets to combat this disease

Response to Reviewer #1: We thank the reviewer for finding our manuscript interesting and appreciating the uniqueness and importance of using 3 different AD models. We agree that "it makes the findings more convincing". We also apologize that the manuscript was difficult to read. The manuscript was substantially revised to improve its presentation style and clarity and, of course, to include your suggestions.

Reviewer #1 commented: "1-In figure 1, since the manuscript does not really show any mechanism that explains the role of B cells in AD, it should at least include better characterization of the dynamics of B cell involvement. It would also be important to show these dynamics and numbers in the PS1APP model since the behavioral testing shown in figures 1F and 1G were done in this model. It would also be important to show the presence of B cells in the brains of these mice, either by histology or by flow cytometry. "

- Our answer: As suggested, we have characterized B cells in our AD mice. The results are now depicted in Fig. 1A-G; Fig.2C; Fig.S1A-J; Fig.S2A-C; Fig.S5A; Fig.S6A-E. We also quantified B cells in the brain parenchyma using immune fluorescent staining and found that AD markedly increases their infiltration in the brain, including hippocampus (please see Fig. 5G and Fig.S7C-E). Moreover, together with the increase of brain B cells, we also detected marked upregulation of IgG in the brain parenchyma and around A β plaques (please see Fig.5H,I). Importantly, upon transient depletion of circulating B cells, the increase of B cells and IgG was lost together with the retardation of AD symptoms in 5xFAD mice (please see Fig. 5G-I and Fig.S7C-E). Overall, these previously unknown results indicate that circulating B cells infiltrate the AD brain, where they presumably exacerbate neuroinflammation. Because transient depletion of circulating B cells ameliorated AD and also eliminated the IgG accumulation in the brain but not in the sera/circulation (please see new data in Fig.S8), we concluded that brain IgG, at least in part, was involved in promoting this disease. As discussed in the revised manuscript (please see reference 52), IgG can exacerbate neuroinflammation, as the disease-associated microglia upregulate Fc γ R and complement in AD - the two AD-associated players that bind IgG.

Reviewer #1 commented: "2-In figure 2, representative images of the whole brain from 3xTg mice and PS1APP mice should be shown as well as quantification of plaques and soluble A β . Showing bits and pieces of data from different mice is not adequate."

- **Our answer:** We have followed the accepted imaging strategy in the field and primarily quantified the subiculum of the hippocampus, as this area is known to be most affected in AD. However, as asked, the manuscript now also contains representative images of the hippocampus (see Fig.S3B-D). The strategy for quantification of A β plaques and activated microglia are described in the Material and Methods section (also see Fig.S3G).

Reviewer #1 commented: "In addition, it is not clear why is the focus only on A β . 3XTg mice also exhibit increased Tau phosphorylation. What happens to Tau as a result of B cell depletion in these mice? This should be easy to test".

- **Our answer:** Although pTau is another important factor, we concentrated on A β plaques as numerous reports from others linked its reduction to the amelioration of the disease. However, as the reviewer asked, we retroactively stained brain tissues from 3xTgAD and B-cell-depleted 3xTgAD mice using antibody for activated Tau (phosphorylated). For unknown to us reasons (possibly due to decay of pTau signals in our tissues after long storage, > 1 year), we failed to detect significant change in pTau (see Fig.1, below). Nevertheless, our data clearly show that the B-cell loss causes significant decrease of the other two factors linked to AD, such as A β plaques and larger/activated microglia.

Figure 1: pTau staining of cryopreserved brain samples from 3xTgAD mice that were pre-depleted or un-depleted of circulating B cells. Note: the tissues were stored for more than one year.

Reviewer #1 commented: " 3-In figure 3, what happens to A β plaque numbers in areas other than the subiculum? The data should be shown."

- **Our answer:** We have extensively checked the hippocampus and, as above-discussed, the major A β plaque accumulation was in the subiculum (see Fig.S3B-D), which was markedly reduced by the loss of B cells. However, in our quantifications, we make sure to gate and quantify A β plaques in comparable areas/locations of the hippocampus.

Reviewer #1 commented: " 4-In figure 4, again data from the whole brain should be shown as well as representative images."

- **Our answer:** As we commented above, the A β plaque accumulation in the hippocampus/subiculum is mostly linked to AD symptoms. We therefore did not stain whole brain and only focused our assays on hippocampus.

Reviewer #1 commented: " 5-In various figures, when the authors refer top large microglia it would be helpful to show representative images so that the reader can have an idea how these microglia look and additional parameters to confirm these are microglia (an not monocytes for example) should be used"

- Our answer: Thank you for your helpful suggestions. The quantification and representative images of microglia are now included in the revised manuscript (see Fig.S3G, C and D). The microglia quantification strategy is described in the Material and Methods section.

Reviewer #1 commented: " 6-A general comment is that deposition of A β and the presence of DAMs are not "symptoms" of AD as the manuscript repeatedly states. They are pathologic manifestations of the disease. This should be corrected throughout the manuscript including abstract."

- Our answer: Revised manuscript is now modified to include the reviewer's point.

=====

Response to Reviewer #2: We thank the reviewer for reviewing our manuscript and properly acknowledging the uniqueness of our study that "This group used 3 mouse models of Alzheimer's disease (AD) to show that B cell deficiency alone prevents AD symptoms. This work suggests that expression of AD transgenes alone is not sufficient to drive disease, it requires B cells. This result suggested that targeting B cells should be considered for the treatment of AD. This was done therapeutically in the mouse AD models, using B cell-depleting agents and it showed efficacy in preventing AD. This work comes at a time when a number of other impactful articles are suggesting a strong involvement of the immune system in AD".

We respectfully disagree with the reviewer's statement that "Previous reports have suggested that the loss of non-specific Ig, which activate microglial phagocytosis of A-beta, is thought to exacerbate AD in some animal models, suggesting a pathogenic role of B cells." The fact that the loss of Ig or B and T cells in Rag-deficient mice exacerbates AD instead indicates that B cells exhibit beneficial/protective, but not AD-exacerbating, roles. Contrary to this potentially protective roles of B cells, our paper for the for the first-time reports that B cells also exhibit pathogenic functions in AD i.e., promote the disease.

Reviewer #2 commented: " Fig 1: Shows increased numbers of B cells that are A4-1BBL+ and IFN-gamma+ in 3xTgAD mice resembling aging B cells in elderly patients but there is no phenotyping of these B cells as B-1a B cells have been shown to also express 4-1BBL. Not clear whether these B cells might be ABC B cells seen in ageing animals (not enough markers were used to distinguish/identify potential B cell subsets.)"

- Our answer: In the revised manuscript, we provide phenotyping results of B cells in AD. They are now depicted in Fig. 1A-G; Fig.2C; Fig.S1A-J; Fig.S2A-C; Fig.S5A; Fig.S6A-E.. As now stated in the discussions, we do not detect involved of ABCs in AD.

Reviewer #2 commented: " The issue with this work is that there is no mechanism. Previous work has shown a role for antibodies. This was not tested here. For instance, what happens when purified Ig from AD mice are injected in AD-BKO mice? I think this is important considering that no B cells were detected in lesions, and a role for antibodies in AD has already been shown. The main question is about the specificity of the B cells expanded in AD. There is little attempt to characterize these B cells and explore the pathogenic mechanism they trigger in disease. In conclusion, as it stands this work may not be novel and just confirm the already described role of antibodies in AD."

- Our answer: We respectfully disagree with the reviewer that "this work may not be novel and just confirm the already described role of antibodies in AD", as there are no known reports that show a pathogenic (AD-promoting) role of B cells in AD.
- To improve the mechanistic insight how B cells might promote AD, the revised manuscript contains new data showing that AD increases B-cell infiltration into the brain parenchyma, including

hippocampus (please see Fig. 5G and Fig.S7C-E). Moreover, together with the increase of brain B cells, we also detected marked upregulation of IgG in the brain parenchyma and around A β plaques (please see Fig.5H,I). Importantly, upon transient depletion of circulating B cells, the increase of B cells and IgG was lost together with the retardation of AD symptoms in 5xFAD mice (please see Fig. 5G-I and Fig.S7C-E). Thus, the increase in IgG and B cells within the brain associates with the amelioration of AD, which is in contrast to the reports from others that IgG reverses the signs of AD. Our results are in concordance with a recent report from others (ref.52) that indicate pathogenic and harmful neuroinflammation-inducing role of brain IgG in MS.

- B cells are not known to be in the brain parenchyma in AD = another novelty of our study. We think that they exacerbate neuroinflammation by, at least in part, IgG because transient depletion of circulating B cells ameliorated AD together with the elimination of B cells and IgG in the brain. Since B-cell depletion did not affect serum/ circulating IgG (please see new data in Fig.S8), we concluded that brain IgG can be involved in promoting this disease. IgG can exacerbate neuroinflammation, as the disease-associated microglia upregulate Fc γ R and complement in AD - the two players that bind IgG.

=====
Response to Reviewer #3: We thank the reviewer finding that " The concept is meritorious especially since CD19/CD20 B-cell depletion is actionable in humans with current FDA-approved drugs in common clinical practice."

Reviewer #3 commented: " The authors highlight the potential neuroprotective benefit of vaccine-induced or naturally occurring antibodies to A β plaques and amyloidogenic peptides in peripheral blood and cerebrospinal fluid of people and mice with AD and further inform the readership that even loss of non-specific immunoglobulins, which activate microglial phagocytosis of A β , is thought to exacerbate AD (ref #20). Despite this relevant preamble, the humoral immune characterization of BKO crossed AD mice is completely lacking. The authors appropriately state that non-specific Igs, Igs with anti-amyloid specificity and B-cells themselves (independently of their ability to make Igs) can affect AD biology. In the original reporting of JHT mice harboring the immunoglobulin JH locus deletion (reference 25), it is clearly stated that these mice lack IgG and IgM and are essentially agammaglobulinemic. Therefore, all the BKO crosses described in this report likely lack Igs as well as B-cells. This experimental approach makes it impossible to separate the effect of absence of Igs from the non-Ig functionalities of B-cells [eg: TNF α production and the like advanced by the authors]."

- Our answer: We thank the reviewer for raising an important issue. We addressed them experimentally and new results are included in the revised manuscript. They now show that AD markedly increases B cells and their product, IgG, in the brain parenchyma, including hippocampus. This increase of B cells and IgG is lost and AD is ameliorated if B cells are transiently depleted from the circulation, indicating a potential pathogenic involvement of B cells and their IgG in this disease. Because the B-cell depletion did not affect levels of IgG in the circulation (serum, Fig.S8), we think that brain IgG caused neuroinflammation by, for example, targeting DAG. Unlike resting microglia, DAG are known to upregulate upregulate Fc γ R and complement in AD - thus would probably respond differently to IgG (see also our arguments in the Discussion section). In addition, as we discussed in the Discussion section, a recent report from others (ref.52) indicate that human brain IgG causes harmful neuroinflammation in MS via targeting microglia.
- As suggested, we tried to either i.v. inject IgG or intra cranially (into hippocampus) in BKO AD mice. Consistent with the report that shows no efficient transfer of IgG into the brain of AD and WT mice (Ref.46), we failed to see any effects of i.v. injected IgG on AD pathology nor on the levels of IgG in B-cell deficient AD mice. In support, unlike genetic B-cell deficiency, the B-cell depletion only eliminated B cells and IgG in the brain without affecting levels of IgG in the circulation. As we

discussed in the Discussions section, these results suggest that brain B cells (which came from the circulation) also locally produced IgG in the brain. Thus, our results contradict to report from Marsh et al (Ref. 21), which linked IgG to the disease protection as it activated microglial phagocytosis of A β plaques.

- Also, we have tried (several times) to reproduce intracranial microinjection of IgG as in Marsh et al (ref.21). However, in our hands, any micro-injection (even saline) into the hippocampus markedly activated microglia. For example, we micro-injected 2 μ l (4 μ g/ μ l) sterile PBS or mouse serum IgG into hippocampus of APP/PS1 mice. After 6 days, mice were euthanized and microglial cells from perfused brains were evaluated by flow cytometrically as well as fluorescent immune staining. Regardless of injected material, microglial cells were comparably activated (see a representative figure, immune fluorescent stained hippocampus, below), questioning usefulness of this-like experiment in revealing IgG-mediated activation of microglia.

Figure 2. Intra hippocampus microinjection activates microglia. Shown are activated microglia (Iba⁺) in the hippocampus of APP/PS1 mice 6 days after intracranial/hippocampal microinjection (shame, PBS or IgG).

Reviewer #3 commented: " The authors further state "Although the molecular mechanism of this process remains unknown and is a topic of a different study, we think that AD induces pathogenic B cells to remotely exacerbate the A β -induced DAM phenotype by inhibiting microglial expression of TGF β 1." This explicit admission highlights the major flaw of this report, namely the descriptive nature of results which do not interrogate two non-overlapping immune functionalities: (i) non-specific Igs and A β -specific Igs from (ii) B-cell function distinct from Ig production. In the genetic crosses used to generate BKOxAD mice, there is no determination whether these mice are hypo or agammaglobulinemic. Assuming these mice lack Igs, is benefit from absence of Igs or absence of non-Ig B-cell functionality?"

- Our answer: As we commented above, we now link AD progression to B cells and their IgG in the brain. Unlike genetic B-cell deficiency (in 3xTgAD-BKO or APP/PS1-BKO mice), transient depletion of B cells did not cause hypo or agammaglobulinemic (see Fig. S8). However, the B-cell depletion was sufficient to eliminate B cells and IgG in the brain and ameliorated AD.

Reviewer #3 commented: " This shortcoming is vexing considering the data contained in Reference #20 which clearly suggested a "protective" role of B-cells in AD pathology in mice [which runs contrary to the hypothesis here pursued]"

- Our answer: To the best of our knowledge, our finding in three different AD models indicate that B cells are required for promoting AD. If we accept that B cells and other immune cells in various pathologies exhibit opposing functions, the AD-promoting feature of B cells presumably does not contradict their beneficial/protective functions. Their pathogenic or protective function would presumably depend on a context, i.e. whether their products (such as IgG or cytokines) encounter resting vs chronically activated microglia. As we briefly discussed in the revised manuscript, the disease associated microglia (at least) upregulate Fc-receptors and complement, the two factors also associated with AD, and thus would differently respond to IgG or its immune complexes.

Reviewer #3 commented: " In another series of experiments mice are treated with B-cell depleting monoclonal antibody, yet the assessment of B-cell depletion on Ig content by this method is not provided. What is Ig content in these mice post treatment? Presumably mature plasma cells in marrow and spleen are unaffected. Is there any measurable anti-amyloid activity remaining in these mice?"

- Our answer: Consistent to our other papers (Bodogai et al., Cancecr Research, 2013; Lee-Chang et al, Blood, 2011; Bodoga et al., Science Tranl. Med, 2018; Ragonnaud et al., Cancer Research, 2019), the anti-CD20/B220 Abs efficiently deplete most B cells in the circulation, except plasma cells. The B-cell depletion result is now shown in Fig. S5A.
- The Ig content and A β -specific Abs data are shown in Fig. S8. Unlike genetic B-cell deficiency, transient depletion of circulating B cells did not affect serum Ig content (Fig. S8). We did not see a tangible A β -specificity in our mice (Fig. S8).

Reviewer #3 commented: "Most substantially, the data presented is descriptive with substantial gaps in mechanism of action of B-cell depletion and we are told this type of work will be performed latterly. This would need to be part of this corpus of work."

- Our answer: The revised manuscript contains additional results that links B cells and IgG in the brain to AD progression. Transient B-cell depletion in the circulation eliminates the increased presence of B cells and IgG in the brains and ameliorates AD.

Reviewer #3 commented: "Minor issue: The treatment protocol of mice with anti-B220 antibody for B-cell depletion is not provided [eg: dosing, frequency] nor is there a reference validating/describing use of this approach"

- Our answer: The missing protocol is now included in the revised manuscript (please see the Material and Methods section). The validation result of the B-cell depletion is shown in Fig. S5A. The protocol was validated in our previous papers, such as Bodogai et al., Cancecr Research, 2013; Lee-Chang et al, Blood, 2011; Bodoga et al., Science Tranl. Med, 2018; Ragonnaud et al., Cancer Research, 2019.

REVIEWERS' COMMENTS

Reviewer #1 (Remarks to the Author):

The revised manuscript addresses most of my concerns and is significantly improved. There are, however, 2 minor concerns that were not adequately addressed. Specifically:

1) In their rebuttal regarding my concerns about figure 4, the authors state:

"• Our answer: As we commented above, the A β plaque accumulation in the hippocampus/subiculum is mostly linked to AD symptoms. We therefore did not stain whole brain and only focused our assays on hippocampus."

This implies that for every data point, the authors only stained the hippocampus. How did they do that, did they dissect the hippocampus from every brain section and stained it separate from the rest of the brain. This is not a convincing response.

2) In their rebuttal regarding my request to show representative images of what they describe as "large microglia" he authors state:

"The quantification and representative images of microglia are now included in the revised manuscript (see Fig.S3G, C and D)."

It is difficult to assess the size of the microglia from figures S3C and D) better quality magnifications should be shown.

Reviewer #3 (Remarks to the Author):

I have reviewed the revised manuscript and companion rebuttal by Authors.

My original assessment that the concept of a pathogenic role of CNS resident B-cells in AD is meritorious especially since CD19/CD20 B-cell depletion is actionable in humans with current FDA-approved drugs in common clinical practice.

In answer to points I had raised, the authors performed additional experiments that links B cells and IgG in the brain to AD progression and transient B-cell depletion in the circulation eliminates the increased presence of B cells and IgG in the brains and ameliorates AD.

This latter information strengthens the authors' closing statement that "the inactivation of B cells can also benefit humans with AD, as therapeutic B-cell removal even at the onset of the disease reversed manifestation of AD in mice."

I had other experimental queries which were not addressed due to entirely believable technical difficulties. However, considering the body of work at hand and the impact of findings and the relative ease with which this concept could be brought to clinic alleviates my concerns.

I had minor editorial comments that have been addressed.

This work provides excellent rationale for a human clinical trial of pharmacological B-cell depletion in early human AD and I hope this study leads to a launch of such.

Authors' point-by-point response to reviewers' comments

We thank all referees of our manuscript for their time and of course, accepting our corrections and answers. Since only reviewer 1 has raised 2 minor concerns, the manuscript was modified to primarily address those comments. Our answers to those comments are as follows:

Reviewer #1 commented: *[The revised manuscript addresses most of my concerns and is significantly improved. There are, however, 2 minor concerns that were not adequately addressed. Specifically:1) In their rebuttal regarding my concerns about figure 4, the authors state: "• Our answer: As we commented above, the Ab plaque accumulation in the hippocampus/subiculum is mostly linked to AD symptoms. We therefore did not stain whole brain and only focused our assays on hippocampus." This implies that for every data point, the authors only stained the hippocampus. How did they do that, did they dissect the hippocampus from every brain section and stained it separate from the rest of the brain. This is not a convincing response.]*

- Our answer: We apologize for our confusing statement and not describing the method properly. We dissected the whole brain and used the left or right half for immunofluorescence staining. Coronal sections were prepared for the brain and the staining was performed for the whole brain slice. Therefore, a new statement was now included in the Results, which states that "Given that in this model the early intra-neuronal A β deposition in the subiculum is linked to cognitive impairments³² and that the subiculum and hippocampal CA1 atrophy is increased in AD patients³³, from hereon we primarily analyzed the subiculum." We also expanded the description of our staining procedure in the Methods section.

Reviewer #1 commented: *[2) In their rebuttal regarding my request to show representative images of what they describe as "large microglia" he authors state: "The quantification and representative images of microglia are now included in the revised manuscript (see Fig.S3G, C and D)." It is difficult to assess the size of the microglia from figures S3C and D) better quality magnifications should be shown.]*

- Our answer: In the revised manuscript, we have changed figures with better quality images to address your comments. Now new images in Fig.S3C, D and Fig.S3G now clearly show the sizes of microglia we quantified.